# Canonical Discriminant Mapping of Origins in Andalusian Black Cattle: Inbreeding and Coancestry Decomposition via Mendelian Sampling Variances and Nodal Ancestor Contributions

**DOI:** 10.3390/ani15121781

**Published:** 2025-06-17

**Authors:** Luis Favian Cartuche Macas, María Esperanza Camacho Vallejo, Antonio González Ariza, José Manuel León Jurado, Juan Vicente Delgado Bermejo, Carmen Marín Navas, Francisco Javier Navas González

**Affiliations:** 1Escuela Superior Politécnica Agropecuaria de Manabí «Manuel Félix López», Calceta 130602, Ecuador; l.cartuchem@gmail.com; 2Department of Genetics, Faculty of Veterinary Sciences, University of Córdoba, 14014 Córdoba, Spain; juanviagr218@gmail.com (J.V.D.B.); carmen95_mn@hotmail.com (C.M.N.); 3Institute of Agricultural Research and Training (IFAPA), Alameda del Obispo, 14005 Córdoba, Spain; mariae.camacho@juntadeandalucia.es; 4Centro Agropecuario Provincial de Córdoba, Diputación Provincial de Córdoba, 14014 Córdoba, Spain; angoarvet@outlook.es (A.G.A.); jomalejur@yahoo.es (J.M.L.J.)

**Keywords:** Negra Andaluza, transhumance, genetic diversity, Iberian breeds, diversity, founder genes, livestock conservation, common nodal ancestors

## Abstract

The Negra Andaluza is a traditional Andalusian cattle breed with deep cultural and historical roots in southern Spain. Although once essential for agricultural labor and timber transport, today, it faces genetic erosion due to declining population size and changes in land use. Our findings show that regions with limited connectivity exhibit signs of genetic isolation, while areas historically linked by transhumant routes retain greater diversity. These patterns highlight the urgent need for targeted conservation strategies that prioritize the protection of genetically rich nuclei, such as those in Córdoba, and reinforce gene flow through controlled breeding programs. Revitalizing traditional livestock practices like transhumance is not only culturally valuable but also essential for maintaining the breed’s genetic variability, adaptability, and long-term survival.

## 1. Introduction

Genetic diversity refers to the total variety of genetic information within a population. It is a measure of how much variation there is in the genetic makeup of individuals within that population. While populations that enjoy high levels of genetic diversity also enjoy an enhanced ability to adapt to changing environments or challenges like diseases, low genetic diversity frameworks can make a population more vulnerable to such challenges [1].

On the other hand, genetic drift refers to the random fluctuation in allele frequencies in a population over time due to chance events, especially in small populations. It is a process that leads to changes in the genetic composition of a population independently of natural selection [2]. Genetic drift can reduce genetic diversity because, over time, some alleles may be lost or become fixed in a population purely by chance, not because they provide any advantage or are selected for.

The genetic diversity of populations, which is closely related to the extent of genetic drift they undergo, has been widely monitored using parameters derived from genealogical data [3]. The most commonly referenced metrics include the average inbreeding coefficient ($\bar{F}$, [4]), which quantifies the probability of an individual inheriting two alleles that are identical by descent; the average coancestry coefficient ($\bar{f\;}$) [5], which measures the genetic relatedness between individuals in a population; and the effective population size, which is directly related to the inbreeding rate per unit of time [6].

Although $\bar{F}$ and $\bar{f\;}$ remain fundamental metrics in population genetics, neither of these parameters explicitly tracks the specific contributions of genes from each ancestor. Therefore, they provide limited insight into how and when specific ancestors shaped the current genetic structure. Hence, they summarize genetic similarity but do not distinguish the underlying sources of homozygosity or identify the historical contributors to genetic drift and, by extension, do not account for the detailed genetic contributions of individual ancestors.

For these reasons, other parameters and methods have been proposed to support the assessment of genetic diversity more comprehensively, including the founder genome equivalent [7], the effective number of ancestors [8], and the effective number of non-founders [9], and the decomposition of ($\bar{F}$) and ($\bar{f}$) into the contributions of nodal common ancestors (NCAs) and Mendelian sampling variances (MSVs).

These approaches allow us to trace the flow of genes across generations, quantify the impact of influential ancestors, and better understand the temporal dynamics of inbreeding—key elements for managing genetic resources in endangered breeds. Particularly, $\bar{F}$ and $\bar{f\;}$ partition can be helpful in understanding who contributed the most to drift and when. Decomposing these values into ancestral components can provide valuable insights into how genetic diversity has evolved over time, identifying which individuals contributed the most to genetic drift and at what point in history [9,10].

Understanding the distribution of genetic diversity across generations is essential for monitoring conservation status, particularly in endangered breeds where historical events, breeding strategies, and demographic shifts have significantly shaped current genetic variability. Probabilities of gene origin derived from pedigree data are valuable tools for monitoring genetic diversity. An analytical relationship has been established between the genetic contributions of nodal common ancestors (NCAs)—those forming inbreeding loops [4], and the contributions of Mendelian sampling variances (MSVs) from ancestors. This relationship demonstrates that the contributions of MSVs to individual inbreeding coefficients, mean inbreeding, or average coancestry can be efficiently converted into the genetic contributions of NCAs and founders, even in large and complex populations.

Wright’s [4] path-counting method provides a detailed breakdown of the genetic contributions of NCAs to individual inbreeding coefficients. However, this method presents a limitation when an NCA is a direct descendant of another NCA, as it only provides a marginal contribution for the older ancestor. To address this, Lacy [11] proposed a decomposition method that attributes inbreeding components to each founder, allowing for a more refined understanding of genetic contributions. Similarly, Caballero and Toro [9] introduced a general framework for partitioning coancestry into contributions from both founders and non-founder ancestors, enabling the estimation of genetic drift components more accurately. The proposed method provides a powerful approach for identifying the most influential sources of homozygosity within extensive and intricately structured pedigrees.

In this framework, Caballero and Toro [9] proposed a general method for partitioning $\bar{f\;}$ into components due to founders and non-founder ancestors. Later, Sargolzaei and Colleau [12], using the aforementioned as a reference, would describe a refined model for decomposing *F*, $\bar{F}$ and $\bar{f}$ into ancestral components through the study of MSVs. This revealed the analytical link between Wright’s path-counting method [6] and the method by Caballero and Toro [9] that enhanced the understanding of historical genetic contributions and drift effects.

The contributions of MSVs of ancestors play a crucial role in assessing genetic diversity and inbreeding within animal populations [12]. MSV refers to the genetic variability that arises due to the random process of gene transmission between generations, which generates inherent variance in the offspring. In pedigree analysis, it is essential to break down an individual’s inbreeding or coancestry level based on the genetic contributions of its ancestors, as these variances provide an accurate measure of how genetic material inherited from previous generations influences the current genetic structure. This approach allows for more detailed identification of the effects of NCAs on the formation of homozygosity and the preservation of genetic diversity, especially in breeds with limited pedigrees or endangered populations. By quantifying MSV contributions, a deeper understanding of how breeding practices and the genetic history of a population impact its sustainability and genetic health can be achieved.

In this framework, canonical discriminant analysis (CDA) arises as an exceptional statistical tool to track NCAs and identify inbreeding loops within pedigrees. By projecting individuals into a reduced multivariate space based on genetic similarity, this technique facilitates the detection of ancestral nodes that disproportionately contribute to inbreeding coefficients and coancestry patterns. Building upon these approaches, this study introduces a novel methodology that decomposes both inbreeding (F) and coancestry (*f*) into distinct ancestral components. Using discriminant analysis, it becomes possible to assign genetic variance to specific historical lineages, offering deeper insights into the origins and propagation of inbreeding within a population.

This statistical framework is applied to the Andalusian Black cattle, a native Spanish breed from the Autonomous Community of Andalusia. This breed, historically used for draft work in agriculture, has undergone significant demographic and functional shifts due to modernization. With the advent of mechanized farming, its primary role transitioned from labor to meat production, leading to a reduction in population size and a geographic concentration in mountainous grazing systems [1]. The breed is currently classified as endangered, with an officially declared population of 1797 individuals distributed across 22 farms as of 2020. Despite ongoing conservation efforts—including the implementation of a pedigree book, controlled breeding programs, and performance monitoring—the Andalusian Black cattle remain at risk of further genetic erosion. Analyzing the genetic structure of this population is therefore essential to ensure its long-term sustainability and to develop informed conservation strategies [1].

One of the key aspects influencing the genetic diversity of the Andalusian Black cattle is its historical connection to transhumance routes, specifically the Cañadas Reales. In parallel, since transhumance is a prehistoric practice for cattle, goats, and sheep, it is tied to native breeds that provide environmental benefits, landscape conservation, and the production of traditional local products with protected designation of origin. This activity is also linked to summer pastures in public mountain areas and winter pastures that are usually privately owned dehesas (Directorate for Sustainable Rural Development, 2012 [13]).

These ancient pathways, used for seasonal livestock migration, played a crucial role in the exchange of genetic material among Iberian breeds. The movement of cattle along these routes not only contributed to the breed’s adaptability to diverse environmental conditions but also influenced the structure of its genetic diversity. By integrating genealogical data with a discriminant analysis approach, this study aims to uncover the historical pathways that shaped the genetic makeup of the breed. This approach enables a more detailed understanding of past genetic bottlenecks, drift events, and the specific ancestors that played a significant role in shaping the current population structure.

The primary objective of this study is to utilize the decomposition of inbreeding and coancestry into ancestral components to identify the key centers of creation, distribution, and propagation of genetic diversity within the Andalusian Black cattle breed. By analyzing genetic variability in relation to the population structure, the study aims to quantify the risk of genetic diversity loss and pinpoint the most critical lineages for conservation. This approach will provide a comprehensive understanding of the genetic flux within the breed, highlighting key genetic reservoirs that should be prioritized in conservation programs. Additionally, the findings will inform more effective conservation strategies, ensuring the maintenance of genetic variability and the long-term sustainability of the breed.

## 2. Materials and Methods

### 2.1. Genetic Analyses

#### 2.1.1. Pedigree Database and Software Tool

The pedigree dataset for the Spanish herdbook of the Black Andalusian cattle breed was provided by the Association of Breeders of the Black Andalusian Cattle Breed (Pozoblanco, Córdoba). Appendix A provides additional official information on the territorial distribution, management systems, reproductive strategies, and conservation efforts currently implemented for the breed.

The complete dataset used in this study encompassed the historical population of the breed, consisting of 8555 animals (both deceased and living), including 3593 bulls and 4962 cows born between January 1994 and May 2020, due to data availability.

In addition to performing demographic and genetic analyses on the entire pedigree (historical population), a subset of the data representing the current population—comprising all living animals—was also analyzed. This subset included 2472 individuals (593 bulls and 1879 cows) born between February 2003 and May 2020.

Genetic diversity calculations, probabilities of gene origin, and founder analyses can only be reliably conducted by including individuals with both parents known or by comparing these individuals to broader historical and current datasets, as recommended by Arias et al. [14] and Navas et al. [15]. Consequently, a third dataset—referred to as the reference population—was established. This reference population included 7351 animals from the current population (3435 bulls and 3916 cows), all of which had both sire and dam identified (i.e., the first generation was fully known). According to Navas et al. [15], using datasets where the parental generation is completely known provides valuable insight into the potential biases in genetic diversity parameters—particularly in endangered breeds, which often have incomplete pedigrees. Detailed information on sample composition is presented in Table 1. The Individual Rate of Coancestry $(\bar{∆C}$) and the degree of assortative mating (α) was calculated using the ENDOG software (version 4.8) [16], a tool specifically developed for analyzing pedigree data to estimate parameters related to genetic diversity, such as inbreeding coefficients, effective population size, and generation intervals. Additionally, CFC software (version 1.0) was employed to calculate nodal ancestral contributions and the probabilities of gene origin. CFC uses path-counting and upward-exploration methods to quantify founder contributions and to decompose parameters such as the inbreeding coefficient (*F*), the average coancestry ($\bar{f}$), and the individual inbreeding coefficient ($\bar{F}$) into contributions of marginal sublineages (MSVs), thereby enabling a detailed assessment of the structure and origin of genetic diversity in the population [17] following the premises in Colleau and Sargolzaei [18].

#### 2.1.2. Individual Rate of Coancestry ($\bar{∆C}$), and Assortative Mating Degree (α)

The individual rate of coancestry ($\bar{∆C}$) for each generation was computed following the methods described in Cervantes et al. [19] through $C_{ba}=1-\sqrt[{\frac{tb+ta}{2}}]{1-C_{ba}}$, where *t_b_* and *t_a_* are the number of equivalent complete generations and *C_ba_* is the *C* for the individuals *b* and *a*. The degree of homogamy, assortative, or non-random mating (α) describes the tendency for individuals with similar phenotypic or genotypic traits to mate more frequently than would be expected under random mating (heterogamy or disassortative mating, where dissimilar individuals are more likely to pair). While assortative mating occurs less frequently than disassortative mating in animal populations, non-random mating patterns still significantly influence genetic structure. Non-random mating was quantified using the method proposed by Caballero and Toro [9], applying the equation;$\;\left(1-\bar{F}\right)=\left(1-\bar{f}\right)\left(1-\alpha \right)$.

#### 2.1.3. Path-Counting and Upward-Exploration Methods

Following Wright’s [4] approach, the inbreeding coefficients of NCAs can be expressed similarly and expanded recursively. Each resulting term represents the genetic contribution of an NCA—regardless of its inbreeding status—to the inbreeding coefficient of the individual in question. NCAs are a subset of common ancestors that appear on both the paternal and maternal sides of an individual’s pedigree. Non-NCAs contribute to inbreeding indirectly through their descendant NCAs. When an NCA is a descendant of another NCA, a portion of the older ancestor’s contribution is captured by the younger one, resulting in only a marginal gene contribution from the older NCA. Afterwards, the upward-exploration method was used to improve computation efficiency. This method is particularly idoneous when dealing with a high number of pedigree paths, as it occurs in the Andalusian Black Cattle.

This tabular method is especially well-suited to partitioning *F*, $\bar{F}$ and $\bar{f}$, across multiple individuals. It begins with a frequency table of labeled gene pairs at the most recent generation. Recursive backward exploration is then applied to trace these gene frequencies to older generations, recalculating frequencies until reaching the founders. Although this method is faster than path counting, it can be memory-intensive for large populations with deep pedigrees, although this is not the situation of the population under study.

To optimize storage, we first decompose *F*, $\bar{F}$ and $\bar{f}$*,* into contributions of MSVs using an indirect method [20], following the foundational concept by Caballero and Toro [9]. These MSV contributions are then used to efficiently replicate the results of both the path-counting and upward-exploration methods.

#### 2.1.4. Decomposing F, $\bar{F}$ and $\bar{f}$ into Contributions of MSVs

To decompose *F*, $\bar{F}$ and $\bar{f}$, we used the method by Colleau [20], who factorized the numerator relationship matrix (A) and exploited its sparsity to indirectly compute average relationships within and between groups, as well as *F*, through matrix–vector multiplication as follows:*A*·*x*

Here, x is a vector containing 1s at the positions of the individual (s) of interest and 0s elsewhere. This computation involves Gaussian elimination of matrices U and UT as follows:A = U^−1^B(U^−1^)^T^

B is a diagonal matrix containing MSVs. Thanks to the structure of U (with −0.5 s linking individuals to their parents and 1s on the diagonal), solutions can be obtained efficiently by tracing pedigrees both up and down.

To obtain the vector m, which holds MSV contributions of ancestors to an individual’s F, consider an individual with sire i and dam j. Setting x = 1 at the positions of i and j, the relationship between them becomes the following:$$F_{j}=\sum_{k}m_{k}$$

Each relationship can be recursively expanded and manipulated algebraically to show that each coefficient equals the proportion of genes each ancestor passed to individual j, consistent with the concept of genetic contribution [21]. Where mk = contribution of MSV of ancestor k to Fj.

For a group of individuals G of size n, this generalizes to the following:$${\bar{F}}_{G}=\frac{1}{n}\sum_{j\in G}F_{j}=\sum_{k}m_{k}$$

To compute m for a group, the indirect method must be run 2 × (number of sires) times: once for the sires and once for their mates. In the first pass, 1s are placed in x at sire positions, and in the second pass at the positions of corresponding dams. This is far more efficient than computing F as self-coancestry (i.e., 1 + F) via recursive pedigree tracing, which would require traversing the pedigree n times. The average coancestry ${\bar{f}}_{G}$ for a group is calculated as follows:$${\bar{f}}_{G}=\frac{1}{n^{2}}x^{T}Ax$$

Again, solving A·x requires only one pedigree tracing pass, contrasting with the more computationally intensive methods. Figure 1 presents a flowchart outlining the decomposition process of the Mendelian Sampling Variance (MSV) used to estimate the contributions of ancestors to inbreeding and coancestry.

#### 2.1.5. Linking Contributions of NCAs and MSVs

For this, let vector u represent the results of the path-counting method for an individual. To trace a single instance of inbreeding, consider a gene sampled from the sire with expected value μ and sampling variance $\sigma^{2}$ = ${MSV}_{i}$, where μ is the genetic value of the sire. If the same gene is sampled twice, the combined expectation is 2μ, the sampling variance is 2σ^2^, and the genetic variance is 2.

The same applies to the maternal gene. Thus, the total genetic variance of the individual includes the following:Genetic variance = MSVi + Paternal variance + Maternal variance

Both paternal and maternal variances can be recursively expressed as weighted sums of MSVs. Let vector mmm contain these contributions. If inbreeding through an individual i occurs with probability fi, then the individual’s contribution to m is:m_i_= f_i_·c_i_

Where c_i_ is the coefficient from the MSV decomposition. The final relationship is as follows:u = C·m

Matrix C is upper triangular with diagonal elements C_ii_ = 1 and C_ij_ = −0.5 if j is a parent of i. Although u = C⋅m, the inverse m = C^−1^u is not easily computable for large populations, Gaussian elimination can be used efficiently due to C’s triangular structure, generating each column of C by tracing the pedigree of the individual’s parents.

Vector u contains the marginal contributions of NCAs to the statistic of interest *F*, $\bar{F}$ and $\bar{f}$, enabling the identification of specific ancestors and time periods associated with inbreeding peaks (e.g., bottlenecks).

To extend this to founder contributions [7,11], let t_jk_ represent the proportion of genes from founder k passed to NCA j. Then, the contribution of founder k via NCA j is as follows:t_jk_⋅u_j_

Summing over all NCAs gives the following direct founder contribution:$$v_{k}=\sum_{j}t_{kj}\cdot \;u_{j}$$

In matrix form, the vector of founder contributions is as follows:v = T^T^⋅uv
where T is the matrix of founder contributions to NCAs. This completes the decomposition from MSVs to NCAs and ultimately to founders.

### 2.2. Statistical Analysis

#### 2.2.1. Canonical Discriminant Analysis (CDA)

##### CDA Methodology

CDA serves as a powerful statistical tool for origin classification. This comprehensive analysis was undertaken to create a robust classification tool capable of discerning intricate patterns within and between ancestor origins based on the information available about them. The location in which every individual was born was considered the dependent variable in the present CDA.
animals-15-01781-t001_Table 1Table 1Definitions of key genetic variables related to inbreeding, coancestry, and mating patterns in structured populations. These concepts are essential for analyzing the contribution of founders and nodal ancestors to genetic diversity and inbreeding dynamics. These are the variables to act as dependent variables in Canonical Discriminant Analysis.Independent VariableDefinitionSourceContribution of nodal ancestors through inbreeding loopsThe proportion of an individual’s inbreeding coefficient that is attributable to nodal ancestors via closed inbreeding paths (inbreeding loops).[17]Contributions of genes of founders to the average inbreeding coefficientThe percentage of the population’s average inbreeding coefficient is derived from the genetic input of the original founders.[17]Contributions of genes of founders to the average coancestryThe percentage of average coancestry between individuals in the population that can be traced back to the genetic contributions of the founders.[17]Coancestry (C)The probability that two alleles, one from each of two individuals, are identical by descent (IBD).[16]Non-Random Mating (α)A mating pattern where individuals do not pair randomly, often based on relatedness, phenotype, or genotype, which affects genetic structure and inbreeding in a population.[16]**Dependent Variables****Definition****Source**Birth MunicipalityMunicipalities where Andalusian Black Cattle individuals were born (Figure 2 and Appendix A) [22]

Table 1 reports the different variables considered in the present CDA as explicative (independent variables). These variables were chosen given that they permit the exploration and decomposition of inbreeding and coancestry through MSVs and the contributions of nodal ancestors. The linear relationship between combinations of the independent variables was considered to determine clustering patterns, considering the birth location as a dependent variable (clustering pattern). The discriminant routine of the Classify package of SPSS version 26.0 software (Armonk, NY, USA, IBM Corp, 2019) and the canonical discriminant analysis routine of the Analyzing Data package of XLSTAT software Pearson Edition 2014 (Addinsoft, Addinsoft, Paris, France, 2014) were used to perform CDA.

Canonical Relationship Plotting

The preliminary phase involved the graphical representation of canonical relationships to spatially distinguish group centroids within the discriminant space. Variable selection was conducted through a regularized forward stepwise multinomial logistic regression procedure, incorporating prior probabilities weighted by group size. This approach aimed to maximize model parsimony and discriminative power in the ensuing CDA. Variables such as nodal ancestor contributions via inbreeding loops and average coancestry were expected to perform well in CDA due to their direct relationship with genetic bottlenecks and founder effects. High nodal contributions often reflect population structuring through repeated inbreeding, while elevated coancestry suggests limited gene flow. Therefore, these variables are particularly suited for distinguishing genetic clusters across birth locations (municipalities and the provinces in which they are located).

2.Sample Size Consideration

A fundamental consideration in the analytical design was adherence to established sample size requirements to ensure statistical robustness. A minimum observation-to-variable ratio of 4:1 to 5:1 was maintained, in alignment with methodological standards that underscore the importance of adequate sample sizes for preserving statistical power and the validity of multivariate inference.

3.Multicollinearity Analysis

To safeguard the validity of the multivariate analyses, a comprehensive assessment of multicollinearity among predictor variables was undertaken. Diagnostics included the Variance Inflation Factor (VIF) and tolerance statistics to quantify the degree of linear dependency. A conservative VIF threshold of 5 was applied to detect and address multicollinearity, thereby preserving the stability and interpretability of the discriminant functions.

4.Canonical Correlation Dimension

The canonical correlation analysis elucidated the multivariate relationships between the predictor set and the grouping variable structure. Canonical dimensions exhibiting correlation coefficients greater than 0.30 were emphasized, as such thresholds denote a non-trivial proportion of shared variance and substantively enhance the interpretive strength of the discriminant model.

5.Discriminant Analysis Efficiency

The efficiency of the discriminant functions was evaluated through the univariate test of group mean equality and Wilks’ Lambda, both serving to assess the contribution and statistical relevance of each predictor. The significance of Wilks’ Lambda was further examined via chi-square approximation (χ^2^), offering critical insight into the extent to which the discriminant functions successfully differentiate among predefined groups.

6.Discriminant Model Reliability

Model robustness was assessed using Pillai’s Trace criterion, which is particularly robust under conditions of unequal group sizes. Statistical significance at the α = 0.05 threshold indicated that the predictor set accounted for meaningful variance across temporal and census-based differentiation among breeds.

7.Canonical Coefficients and Loading Interpretation

Interpretation of the canonical discriminant structure was informed by both standardized canonical coefficients and structure matrix loadings. Variables with absolute loading values ≥ 0.40 were considered substantial contributors to group separation. A stepwise variable selection algorithm was employed to refine the model by excluding predictors lacking statistical significance, thereby enhancing parsimony and interpretability.

8.Discriminant Function Validation and Cross-Validation

Model generalizability was tested through a leave-one-out cross-validation (LOOCV) procedure to evaluate classification accuracy. Press’s Q statistic was used to compare the observed classification rate against a critical chi-square value (χ^2^ = 6.63), establishing whether the model’s predictive accuracy significantly exceeded chance expectations by at least 25%. The LOOCV approach also assessed the likelihood that an individual of unknown origin aligned with the discriminant profile of a leisure/pet-use breed or one maintaining traditional functional roles (e.g., hunting, ratting, shepherding, or guarding). Classification success was quantified by the hit ratio—computed as the proportion of correctly classified cases based on proximity to group centroids—which served as a key metric of model validity and predictive precision.

9.Mahalanobis Distances

Mahalanobis distances were employed to construct a dendrogram that graphically represents the landscape of breeding centers and geographic origins, highlighting provinces and municipalities acting as focal points in the development and dissemination of the breed.

## 3. Results

### 3.1. Descriptive Statistics and Frequency Analysis

Descriptive statistics (Table 2) were calculated for the numerical variables in the dataset, including the contribution of nodal ancestors, founder gene contributions, coancestry, and non-random mating coefficients. The results show a wide range of values, with some variables, such as the contribution of nodal ancestors through inbreeding loops, reaching a maximum of 0.0447 and a standard deviation of 0.0033, indicating considerable variation among individuals. Figure 3 represents contributions of MSVs mean, standard error of the mean (SEM), maximum, and mínimum across birth provinces and municipalities in the Andalusian Black Cattle Breed.

In terms of categorical data, the frequency analysis (Figure 4) revealed that the province with the highest number of records is Córdoba, accounting for 447 entries, followed by Huelva (252 entries), Seville (191 entries), and Jaén (19 entries). These frequencies highlight a significant geographic concentration of data, with Córdoba representing the majority of the sample.

### 3.2. Canonical Discriminant Analysis

#### 3.2.1. Multicollinearity Analysis

In evaluating the reliability of the CDA model, a thorough examination of multicollinearity was conducted. The statistical assessment involved tolerance and VIF values for the independent variables in the analysis. Notably, the contributions of the genes of the founders to the $\bar{F}$, $\bar{f}$, and non-random mating (α) exhibited VIF values above the threshold of 5, indicating multicollinearity. These variables were therefore excluded to preserve model robustness. Their removal also enhances interpretability by reducing redundancy among metrics that reflect closely related population structure features, for instance, those based upon the relationship of $\bar{F}$ and $\bar{f}$ with α through the following equation $\left(1-\bar{F})=(1-\bar{f})(1-\alpha \right)$. This statistical procedure ensures clearer differentiation between variables that uniquely contribute to understanding genetic clustering.

#### 3.2.2. Model Reliability and Explanatory Potential

The Wilks’ Lambda test, using Rao’s approximation, yielded a Lambda value of 0.9806, indicating a high degree of similarity among the groups. However, the observed F-value of 1.9764 exceeded the critical value of 1.8841, and the associated *p*-value of 0.0383 falls below the conventional alpha level of 0.05. These results suggest that there is a statistically significant difference among the groups under analysis. While the Lambda value is relatively high—implying that most of the variance is still unexplained by group differences—the *p*-value indicates that the discriminant function detects meaningful, though modest, differentiation among the groups. This finding supports the relevance of further analysis to interpret the nature and strength of the group separation.

#### 3.2.3. Analysis Efficiency

The results of the discriminant analysis reveal three discriminant functions (F1, F2, and F3), with the first function (F1) contributing the most to group separation. F1 has an eigenvalue of 0.0157 and accounts for 79.72% of the total discrimination, followed by F2 with 19.04% and F3 with only 1.25%. The cumulative discrimination percentage reaches 100% across the three functions, with F1 capturing the majority of the variability.

To assess the significance of these discriminant functions, Bartlett’s test for eigenvalue significance was applied. The test yielded a statistically significant result for F1 (Bartlett’s statistic = 17.7442, *p*-value = 0.0383), indicating that this function contributes meaningfully to group differentiation. In contrast, F2 and F3 showed non-significant *p*-values (0.4604 and 0.6369, respectively), suggesting limited discriminatory relevance.

Taken together, these findings indicate that while multiple discriminant functions were extracted, only F1 significantly contributes to explaining variability among the groups. This supports the validity and usefulness of F1 as the primary source of discrimination in the dataset.

#### 3.2.4. Discriminant Potential

The unidimensional test of equality of means offers valuable insight into the discriminative power of individual variables within the framework of the discriminant analysis. In this case, the variable Contribution of nodal ancestors through inbreeding loops yielded a Wilks’ Lambda value of 0.9942, indicating very limited but still present discriminatory ability. The corresponding F-statistic of 1.7788 and a *p*-value of 0.0500—exactly at the conventional significance threshold—suggest that this variable marginally contributes to differentiating between the predefined groups. Similarly, the variable Coancestry (C) also reached a *p*-value of 0.0500, reinforcing its borderline statistical significance in distinguishing among the groups. Although both variables exhibit high Lambda values, which typically imply weak discriminatory power, their statistical significance implies they still play a relevant role in the classification process. These findings emphasize the nuanced but meaningful contribution of these genetic parameters in characterizing the structure of the groups.

#### 3.2.5. Pillai’s Trace Test

The Pillai’s Trace test was conducted to evaluate the multivariate effect of the grouping variable on the set of dependent variables. The test yielded a Pillai’s trace value of 0.0194, which indicates that approximately 1.94% of the variance in the dependent variables is explained by group differences. The observed F-value of 1.9726 exceeds the critical F-value of 1.8833, with degrees of freedom DF1 = 9 and DF2 = 2724. Importantly, the *p*-value of 0.0386 is below the alpha level of 0.05, indicating that the group differences are statistically significant. These results suggest that, although the effect size is small (as indicated by the low trace value), there is a statistically significant multivariate difference among the groups. Therefore, the hypothesis that group means are equal across the set of dependent variables is rejected in favor of the alternative hypothesis.

#### 3.2.6. Discriminant Coefficients and Classification Patterns

The results for F1 reveal important insights into the genetic structure of the population. The intercept value of −0.2894 indicates a baseline measurement for the model, suggesting a slight negative influence when no other factors are considered. The “Contribution of nodal ancestors through inbreeding loops” is significantly high at 317.1545, highlighting the substantial impact of inbreeding loops and the genetic contribution from nodal ancestors in shaping the population’s genetic makeup. This suggests that the presence of ancestral nodes plays a critical role in the genetic relatedness within the studied group. Furthermore, the coancestry value (C) of 0.0000 indicates no measurable genetic correlation between individuals in the population, suggesting that there is no observable increase in relatedness because of inbreeding, at least within the context of this specific analysis. These results provide a nuanced understanding of how genetic factors, particularly those related to inbreeding loops, influence the population’s genetic structure in the studied context.

#### 3.2.7. Discriminant Function Validation and Cross-Validation

The generalizability of the model was assessed through a LOOCV procedure to evaluate its classification accuracy across multiple regions. The results showed that while the model performed exceptionally well in classifying cases from Córdoba, with a high accuracy of 99.55%, it struggled significantly with other regions, such as Huelva, Jaén, and Sevilla, achieving an overall accuracy of only 49.12%. Specifically, regions like Huelva and Jaén had 0% accuracy, suggesting that the model could not differentiate these cases effectively from other categories. This indicates the need for further refinement in the model, such as better feature selection or adjustments in the classification approach, to improve predictive precision for regions with lower classification success. Press’ Q statistic was used to compare observed classification accuracy against chance expectations, confirming that the model performs better than random chance, but there remains room for improvement, especially in distinguishing between regions with similar profiles. Prior and posterior classification for ancestors across provinces and municipalities are shown in Figure 5 and Figure 6, respectively.

#### 3.2.8. Mahalanobis Distances

The Mahalanobis distance matrix was computed to assess the similarity across the different municipalities and provinces (CO, Córdoba; HU, Huelva; SE, Sevilla) considered in the study based on a multivariate distribution of the genetic diversity parameters that were evaluated. The diagonal elements of the matrix, which represent the distance of each location (municipality and province between parentheses) to itself, all returned a value of zero, as expected. Off-diagonal distances provide insight into the relative similarity between locations, with smaller values indicating greater similarity. For example, locations like Alajar (HU) and Alcalá de Guadaira (SE) exhibited a very low distance of 0.002, suggesting that these locations are highly similar in the context of the measured variables. In contrast, the distance between Alajar (HU) and Constantina (SE) was much higher (0.479), reflecting a more distinct difference between these two locations. Additionally, several locations, such as Zufre (HU), displayed multiple near-zero distances to other locations, indicating high similarity across several sites. The analysis revealed notable clusters of locations with similar characteristics, while other locations, such as Espiel (CO) and Guillena (SE), showed greater dissimilarity from others. These results suggest that while some locations share common traits, others are distinguishable based on the multivariate features considered in the analysis, highlighting the diverse environmental or cultural factors influencing the grouping of these locations. Figure 7 depicts a representation of Mahalanobis distances across Andalusian Cattle Breed Municipalities.

## 4. Discussion

Our results contribute to tracing the historical evidence surrounding the origin and evolution of the Andalusian Black cattle breed. Rooted in ancient bovine lineages, the Andalusian Black cattle reflect a blend of Near Eastern, Italian, and North African genetic influences. While this background provides essential context, its key implication lies in the breed’s resilience and adaptation across diverse Iberian terrains, traits that are mirrored in its retained diversity despite modern demographic pressures. These early domesticates gradually spread across Europe, where regional selection pressures and breeding practices led to the development of the diverse cattle breeds observed today [23].

In the Iberian Peninsula, the so-called “Iberian trunk” is derived from local aurochs populations. This becomes particularly evident in breeds that are typically characterized by dark coat colors, including black, grey (*cárdena*), and black-and-white or red-and-white (*berrenda*) varieties [24]. These breeds adapted to the granitic–siliceous substrates and calcium-poor pastures typical of central and western Iberia, with a historical presence notably concentrated in regions such as the Sistema Central and La Rioja (Figure 8).

Breeds from the Iberian trunk, such as Serrana de Teruel, Serrana de Soria, Avileña, Preta Portuguesa, Morucha, Sayaguesa, Caldelá, Cárdena Andaluza, and Andalusian Black (Negra Andaluza) have played an essential role in the development of regional livestock systems. These breeds are traditionally found in the mountainous areas of La Rioja, Soria, Teruel, and the Central System, regions characterized by significant territorial mobility that extended into southern territories such as Extremadura and the Alcudia Valley. Historically, these cattle were used for hauling timber from the pine forests of the Central System and other Iberian mountain ranges—a role that earned them the designation *pinariegas* cattle—and contributed to their dispersal across Andalusia [1].

The use of these breeds as draft animals by the Royal Guild of Carters (Real Cabaña de Carreteros) further facilitated their expansion southward, particularly into the Tagus Valley (Figure 8 and Figure 9). Although populations of this lineage likely existed since ancient times in the Portuguese foothills of the Central System (Serra da Estrela), their expansion southward and westward was undoubtedly intensified by their role as draught animals in agricultural contexts. With the mechanization of agriculture, these functions have become obsolete, and today, these breeds are reared extensively for their meat production potential [26,27].

Among their former uses, dragging and hauling timber to sawmills and transporting wood to markets have been widely documented. This historical use played a key role in the emergence of the Andalusian Black cattle breed, particularly in regions such as Córdoba and Seville, where gene flow resulting from transhumant movements contributed to shaping the breed’s genetic architecture. Selection for traits such as size, strength, and docility was essential to meet the demands of draft work. The prominent role of these cattle within the Royal Guild of Carters further reinforced their geographical spread and consolidation [1].

One hypothesis suggests that the formation of the Andalusian Black breed in the 19th century was a response to increasing demand for draft animals during the transition from livestock-based to agricultural land use. This transition created a pressing need for *yuntas* (draft pairs) for agricultural labor. To meet this demand, animals from provinces such as Ávila and Salamanca were introduced, leading to crossbreeding processes that gave rise to distinct lines, such as the *Negra de las Campiñas* and the *Cárdena Andaluza*. This period of agricultural expansion was instrumental in consolidating the genetic basis of the Andalusian Black breed. Furthermore, the practice of transhumance—particularly along livestock routes (*cañadas*, *veredas*, and *cordeles*)—further influenced both the genetic diversity and geographical distribution of the breed [1,28,29,30,31].

Transhumance, a prehistoric practice associated with the seasonal movement of bovine, caprine, and ovine species, represents a cornerstone of the Andalusian Black cattle’s genetic heritage. These movements, connected to public-use mountain areas and privately managed *dehesas*, have played a fundamental role in shaping both the spatial distribution and genetic diversity of the breed. The main transhumance areas in Andalusia include the Sierra Cordobesa, Sierra de Aracena, and the Guadalquivir Valley—regions that today remain crucial reservoirs of genetic diversity for this breed [1,32].

The core distribution area of the Andalusian Black cattle breed extends from the Sierra of Córdoba to the Sierra de Aracena, also encompassing zones such as the northern Sierra of Seville, the Valle de Los Pedroches and the Valle del Guadiato. According to Columela [33], traditional Andalusian cattle included three main types: *Morena*, *Roja,* and *Rubia*. The *Morena*, described as eumetric and brevilinear, was considered representative of the fine *casta* cattle and was frequently used in crossbreeding to improve temperament and enhance functional traits in working animals. Its compact body—marked by a voluminous trunk, shorter and finer limbs and head, and a lighter skeleton—also made it advantageous for meat production, reinforcing its agricultural and economic value [32].

From the crossing of the *Morena*, *Roja*, and *Rubia* types, several local varieties emerged, often quite similar in phenotype. However, breeder selection for straight, black-coated, and compact sires led to the emergence of a relatively uniform type: the *Negra Andaluza*. This process was further supported by the incorporation of *Avileña Negra Ibérica* bulls as improvers. Historically known as *Negra Campiñesa* or *Negra de las Campiñas*, the breed was thus distinguished from the *Avileña Negra Ibérica*, more common in the *dehesas* of Extremadura and Castile [33].

The resulting population, characterized by black coats and functional conformation, established itself mainly in the southern provinces of Córdoba and Seville, where the breed’s genetic base consolidated.

The genetic structure of the *Negra Andaluza* has been shaped by founder effects, inbreeding loops, and the influence of nodal ancestors. Analyses indicate that Córdoba is the primary genetic nucleus, with a significantly higher number of recorded ancestors (448) compared to Huelva (252) and Seville (193). This concentration of genealogical contributions suggests that Córdoba played a central role in shaping the breed’s genetic architecture.

Despite its broad range, two major zones of concentration can be identified: one in the Sierra Morena and its foothills (Córdoba and Huelva) and another in the lowlands of Seville, Cádiz, and Huelva. As shown in Figure 4, the highest nodal contributions through inbreeding loops are concentrated in Córdoba. Similarly, the greatest contributions of founder genes to both average inbreeding and coancestry coefficients have been observed in Córdoba and Seville, as reported by Cartuche Macas et al. [1] using Wright’s parameters. These findings support the hypothesis that the *Cañada Real Soriana Oriental* was a major axis for the genetic establishment of the breed in the region, a process reinforced by Andalusia’s livestock trails (*cordeles*, *veredas*, and *coladas*; see Figure 9).

Within Córdoba, the municipality of Cabra stands out for its exceptionally high contribution of nodal ancestors and founder genes, both to inbreeding and coancestry (Figure 5). This prominence may be due to its relative geographical isolation in the southern part of the province, with limited connectivity to other nuclei, which would have restricted gene flow and led to reduced genetic diversity. In contrast, areas better connected through transhumance routes (*cañadas* and *cordeles*) exhibit greater genetic exchange and heterogeneity. Similarly, the Hinojos nucleus in Huelva, due to its geographical isolation, exhibited a greater contribution to both inbreeding and coancestry, underscoring the critical role of isolation in shaping the genetic makeup of Andalusian Black Cattle [32]. The influence of nodal ancestors is key to understanding regional differentiation, especially in Córdoba, Seville, and Huelva, where historic livestock routes (cañadas, veredas, cordeles, and coladas) facilitated gene flow across the region [1].

Notably, the Cañada Real Soriana Oriental linked Córdoba and Seville, promoting the spread of founder genes that significantly shaped the breed’s current genetic structure. Conversely, Huelva’s connection through the Cañada Real Soriana Occidental and the Cañada Real de la Plata—linked to the Leonesa Occidental and Oriental and Segoviana routes—allowed for a more limited but still meaningful genetic exchange. The relatively low genetic input from Jaén is likely due to its weak connection via the Real Conquense and Soriana Oriental routes [1,28,30,31].

Regarding the number of identified ancestors, Jaén recorded the fewest (19), while Córdoba had the most (448), followed by Huelva (252) and Seville (193). This underscores Córdoba’s genetic prominence, with more than double the number of ancestral contributions compared to Huelva and Seville. Wright’s parameters for coancestry and inbreeding further confirm Córdoba as the primary center of genetic consolidation, reinforcing its pivotal role in the development of the breed’s distinct genetic traits.

The origin and structure of Andalusian Black Cattle can be traced through inbreeding loops, MSVs, and contributions from nodal ancestors. CDA was employed to evaluate how these variables—essential components of genetic architecture—contribute to population differentiation across Andalusia [1].

Multicollinearity was observed among variables such as the founder contributions to average inbreeding, average coancestry, and non-random mating (α), along with the contribution of nodal ancestors through inbreeding loops and coancestry (C). These interdependencies, both conceptual and mathematical, reflect overlapping aspects of population structure and mating patterns. To ensure model robustness and interpretability, collinear variables were excluded, retaining only those offering distinct, non-redundant information.

The analysis revealed statistically significant yet modest differences between groups, with the first discriminant function (F1) emerging as the most informative. Inbreeding loops and nodal ancestor contributions were central to explaining group structure.

Previous studies have established that domestic cattle, including Andalusian Black Cattle, descend from the extinct Eurasian aurochs, with the Iberian Peninsula acting as a major center for the development of native breeds. The Andalusian Black is thought to have originated from historical breeding events in the fertile regions of Andalucía, influenced by the introgression of genetic diversity from areas like Ávila and Salamanca [1,23,34,35].

Inbreeding and coancestry are central to the breed’s genetic architecture, with inbreeding loops playing a significant role in maintaining specific traits. Our analysis showed that MSVs enabled a quantitative decomposition of inbreeding, highlighting the critical contribution of nodal ancestors—particularly in Córdoba—where these loops were more prevalent and impactful [28,36,37,38,39].

The regional variation in the contribution to inbreeding was most pronounced in Córdoba, suggesting a potential genetic bottleneck due to limited gene flow. Although the analysis revealed pronounced inbreeding loops, the breed appears to have retained a moderate level of genetic diversity, as indicated by the coancestry mean of 0.000475 and inbreeding loop contribution mean of 0.000949 (Table 2). This resilience may be attributed to historical transhumance, which enabled genetic exchange across regions despite isolation (see Figure 9). Previous work by Cartuche Macas et al. [1] and Navas González et al. [15] also supports this finding, highlighting the buffering effect of migratory breeding on genetic erosion.

The observed discrepancy between inbreeding and coancestry may be attributed to the complex interplay of genetic drift, selection, and gene flow. While inbreeding loops can lead to the fixation of alleles, connectivity through historical transhumance routes likely prevented excessive genetic homogenization [35,37].

The primary driver of inbreeding in Andalusian Black Cattle appears to be the influence of nodal ancestors. CDA revealed their disproportionate impact on the breed’s genetic structure, particularly in Córdoba and Huelva, where their contributions to inbreeding and coancestry exceeded expectations. These findings suggest the persistence of distinct genetic lineages tied to specific founders.

Wright’s parameters further corroborated Córdoba’s strong genetic signature, characterized by a higher proportion of ancestral contributions to inbreeding loops and coancestry. LOOCV confirmed this pattern, with Córdoba achieving high classification accuracy, while Huelva and Seville showed lower discriminant precision. This emphasizes Córdoba’s genetic isolation and its central role in shaping the breed’s diversity [1,35,36].

The geographic concentration of the breed and the clear regional variation in genetic composition reflect distinct selection pressures. Córdoba and Huelva exhibited higher relatedness and fewer external genetic inputs, whereas Seville displayed a more homogenized profile, likely facilitated by greater gene flow through transhumant livestock routes.

The model’s inability to classify individuals from Huelva and Jaén may be due to several factors. First, limited sample sizes (especially in Jaén, with only 19 individuals) reduce classification accuracy. Second, sampling bias may exist if animals from these regions are underrepresented in pedigree completeness or geographic coverage. Additionally, genetic homogeneity—either from bottlenecks or founder effects—could obscure distinctiveness in the discriminant space. These issues suggest a need for enhanced sampling, particularly in these provinces, and potentially incorporating molecular markers to uncover hidden genetic structures.

CDA identified three discriminant functions, with the first function (F1) explaining the vast majority of the total variance (79.72%). This dominance suggests that the primary axis of genetic differentiation among the groups is strongly associated with the variables contributing most to F1. Notably, this function was chiefly influenced by inbreeding loops and the genetic contributions of nodal ancestors, highlighting their central role in shaping the observed genetic structure.

The secondary (F2) and tertiary (F3) functions contributed comparatively less to the overall variance, implying that while they capture additional dimensions of genetic differentiation, their explanatory power is limited. These functions may still reflect biologically meaningful patterns, potentially related to more subtle demographic or historical factors, but they do not overshadow the primary signal associated with inbreeding and key ancestral contributions.

Interestingly, while coancestry was also included in the analysis, its influence appears secondary. This suggests that, in the studied population, patterns of genetic differentiation are driven more by internal lineage structures (e.g., repeated inbreeding and strong founder effects from nodal ancestors) than by broader patterns of relatedness among individuals or subpopulations. In practical terms, this underscores the importance of managing inbreeding and monitoring influential ancestors in conservation or breeding programs, as these factors may have a disproportionate effect on genetic diversity and population structure [1,35,37].

The prominence of nodal ancestors highlights the historical influence of selective breeding, where specific individuals or family lines were favored for traits such as size, temperament, or adaptation to local environments. These past practices have left a lasting genetic legacy in the Andalusian Black Cattle.

In conservation terms, the breed has maintained high genetic diversity despite the influence of inbreeding. Historical gene flow across extensive livestock routes likely played a crucial role in preventing significant genetic erosion. Nonetheless, the continued presence of inbreeding loops signals the need for future breeding programs to manage genetic diversity carefully and mitigate risks associated with further inbreeding.

## 5. Conclusions

This study reveals that the origins and genetic structure of the breed are profoundly shaped by inbreeding, coancestry, and the influence of key nodal ancestors. While inbreeding loops and maximum selfing values (MSVs) have significantly contributed to the breed’s genetic makeup, gene flow between subpopulations has played a crucial role in maintaining genetic diversity. These findings highlight the ongoing tension between historical breeding practices and the preservation of genetic variability, particularly in geographically isolated populations. Although the discriminant model demonstrated high classification accuracy for certain regions, its variable performance across others indicates the need for further refinement to better represent regional genetic distinctions. Importantly, this study introduces a novel methodological framework that integrates genealogical decomposition with canonical discriminant analysis, offering a powerful tool for evaluating genetic structure and informing conservation strategies. This approach is not only applicable to the conservation of Andalusian Black cattle but also holds significant potential for broader application to other endangered autochthonous breeds facing similar genetic erosion due to demographic contraction and genetic drift. By providing both analytical depth and practical utility, this framework contributes meaningfully to the field of conservation genetics.

## Figures and Tables

**Figure 1 animals-15-01781-f001:**
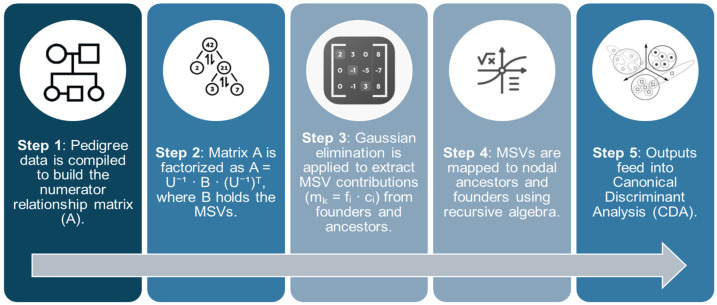
Flowchart of the MSV decomposition process for estimating ancestor contributions to inbreeding and coancestry, based on matrix factorization and recursive pedigree tracing.

**Figure 2 animals-15-01781-f002:**
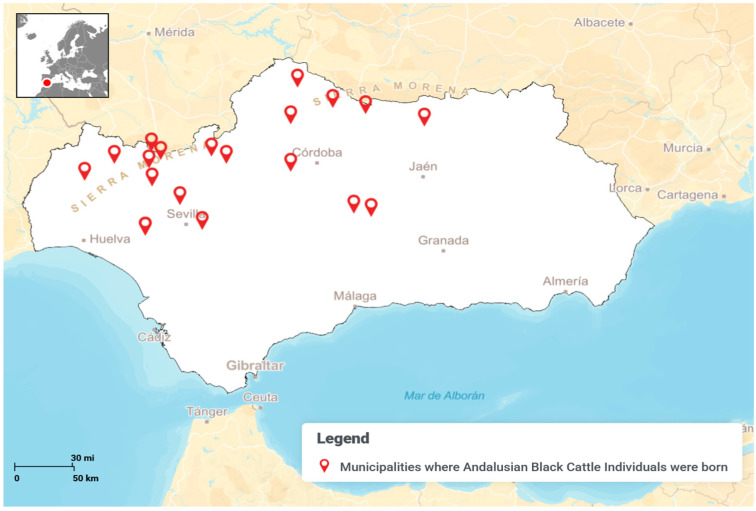
Map of Andalusia (Spain) depicting municipalities where Andalusian Black Cattle individuals were born.

**Figure 3 animals-15-01781-f003:**
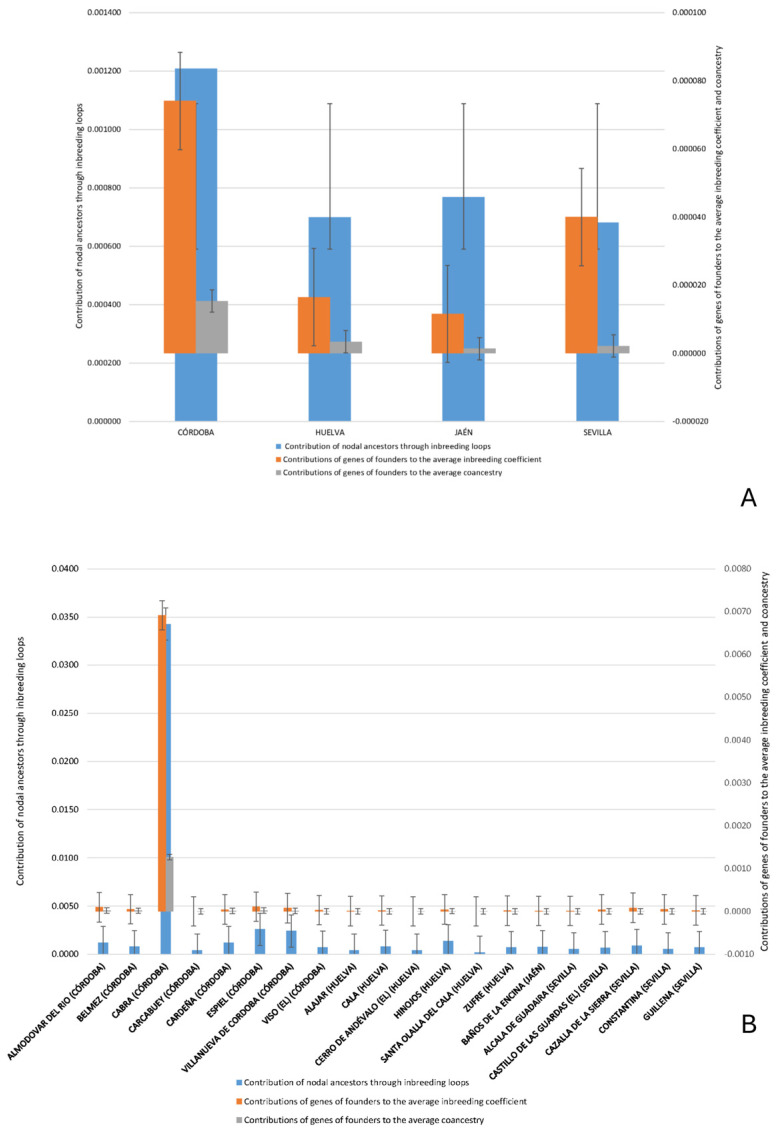
Contributions of Mendelian sampling variances (MSVs) mean, standard error of the mean (SEM), maximum and mínimum across birth provinces (**A**) municipalities (**B**) for Andalusian Black Cattle Breed.

**Figure 4 animals-15-01781-f004:**
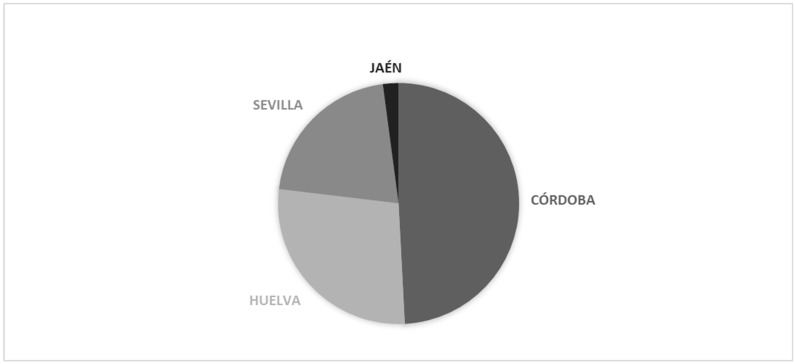
Distribution of Black Cattle Breed individuals across Andalusian Provinces.

**Figure 5 animals-15-01781-f005:**
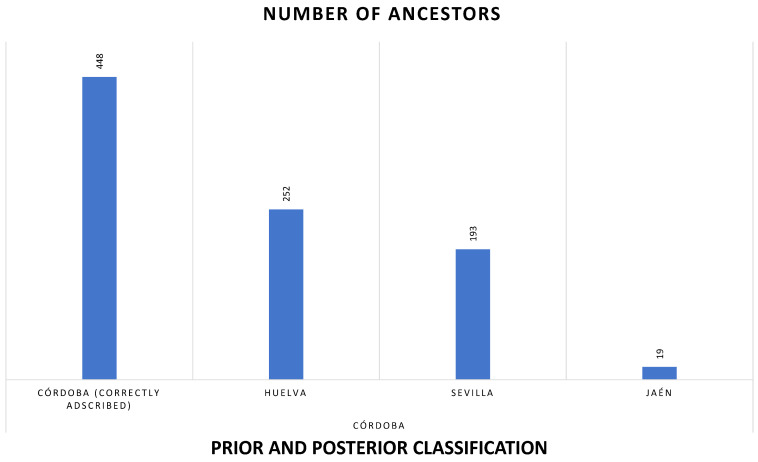
Prior and posterior classification of Cordobesian ancestors across the provinces where Andalusian Black Cattle were born, with Córdoba being the predominant birth province, therefore taken as a reference.

**Figure 6 animals-15-01781-f006:**
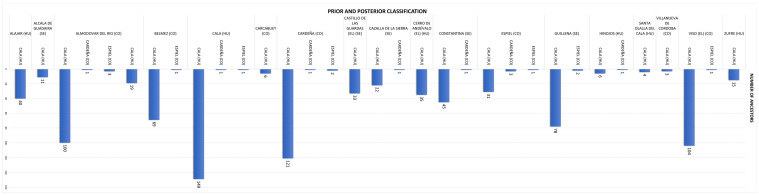
Prior (vertical labelling) and Posterior (horizontal labelling) classification of ancestors across the different municipalities and provinces (CO, Córdoba; HU, Huelva; SE, Sevilla) in which the Andalusian Black Cattle was born.

**Figure 7 animals-15-01781-f007:**
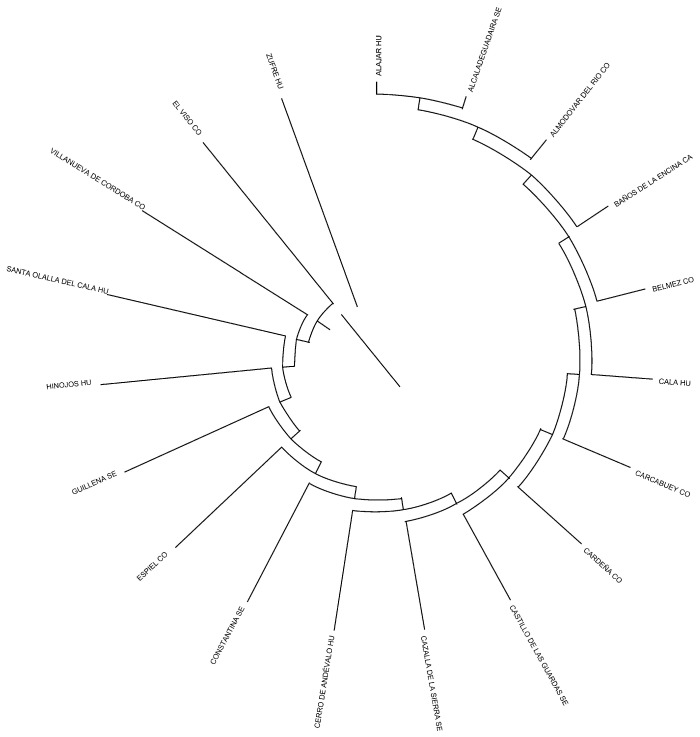
Mahalanobis distances dendrogram depicting relationship across municipalities and provinces (CO, Córdoba; HU, Huelva; SE, Sevilla) in the Andalusian Black Cattle Breed.

**Figure 8 animals-15-01781-f008:**
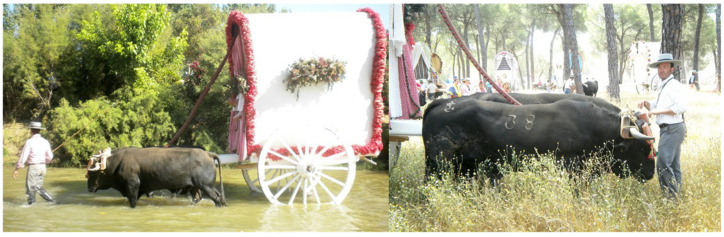
Andalusian Black Cattle in their role as a work animal (Ana Jiménez, 2011 [25]).

**Figure 9 animals-15-01781-f009:**
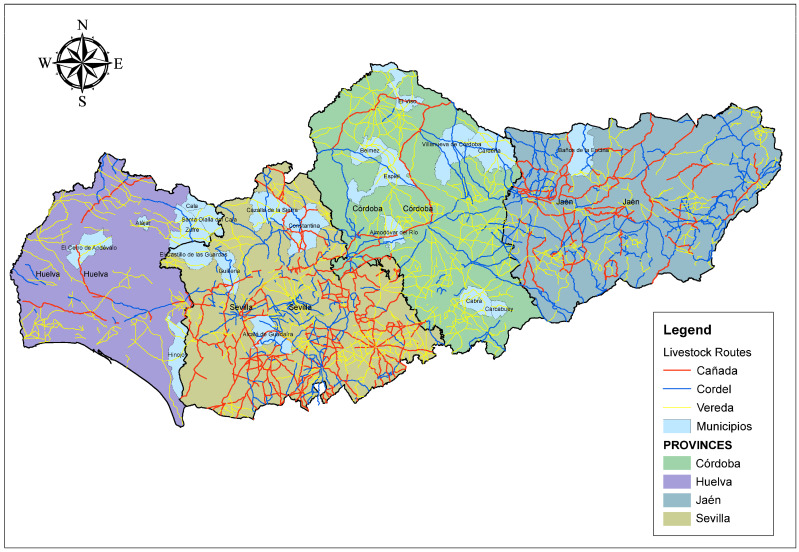
Livestock routes across the area of expansion of the Andalusian Black cattle breed.

**Table 2 animals-15-01781-t002:** Descriptive statistics for independent variables used in the analysis.

Variable	Mean	Min	Max	Standard Deviation
Contribution of nodal ancestors through inbreeding loops	0.000949	0.000117	0.044711	0.003338
Contributions of genes of founders to the average inbreeding coefficient	0.000049	0.000000	0.006913	0.000396
Contributions of genes of founders to the average coancestry	0.000009	0.000000007	0.001270	0.000082
Coancestry (C)	0.000475	0.000058	0.022355	0.001669
Non-Random Mating (α)	−0.000478	−0.022867	−0.000058	0.001697

## Data Availability

Data will be made available from the corresponding author upon reasonable request.

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
