# Peer review of "Canonical Discriminant Mapping of Origins in Andalusian Black Cattle: Inbreeding and Coancestry Decomposition via Mendelian Sampling Variances and Nodal Ancestor Contributions"

_animals, 2025, doi:10.3390/ani15121781_

Round 1

Reviewer 1 Report

Comments and Suggestions for Authors

Line 76. The explanation of what F and f represent should be clearer, especially for readers without a technical background.

Lines, 92, 95 and 128, and all other cases. It would be advisable to abbreviate 'nodal common ancestors' to maintain consistency with the rest of the text, which uses acronyms.

Lines 137–166. It is necessary to include the corresponding bibliographic references for these paragraphs. Although the content is widely known in the Spanish context, I believe that including the relevant citations could improve the text for international readers who wish to deepen their understanding of the topic.

Line 232. This sentence must be corrected: When a NCA ...

Line 234. A space should be added in 'improvecomputation' to separate it into 'improve computation'.

Line 246. The phrase "Mendelian sampling variances" should be removed, leaving only "MSVs," since the meaning of the abbreviation has already been explained earlier.

Line 275. Final point should be added.

Table 1. Since tables should be understandable on their own without requiring information from the main text, it would be better to write out CDA in its full form as Canonical Discriminant Analysis.

Table 1. I suppose it should be 'Figure 1 and Supplementary Table S1'.

Figure 1 should be corrected in this sense: Figure 1. Map of Andalusia (Spain) ...

Line 341.  It would be advisable to abbreviate 'Mendelian Sampling Variances' to maintain consistency with the rest of the text, which uses acronyms. 

Line 346, 348 and 356. It would be advisable to abbreviate 'canonical discriminant analysis' to maintain consistency with the rest of the text, which uses acronyms. 

Line 427. The phrase "Mendelian sampling variances" should be removed, leaving only "MSVs," since the meaning of the abbreviation has already been explained earlier.

Figure 3. The title for Figure 3 is missing and should be added below the figure.

Line 448. The phrase "Variance Inflation Factor" should be removed, leaving only "VIF," since the meaning of the abbreviation has already been explained earlier.

Line 526. The phrase "leave-one-out cross-validation" should be removed, leaving only "LOOCV," since the meaning of the abbreviation has already been explained earlier.

Figure 4 and 5. In my opinion, it is not clear which part represents the prior classification and which represents the posterior classification.

Figure 5. The figure appears pixelated; I believe it would be advisable to modify it to improve the resolution and enlarge the text to enhance its readability and overall clarity.

Line 552. I believe this is the first time the abbreviation HU appears, so it should be written as Huelva (HU).

Line 552. I believe this is the first time the abbreviation SE appears, so it should be written as Seville (SE).

Line 559. I believe this is the first time the abbreviation CO appears, so it should be written as Córdoba (CO).

Figure 6. The meaning of the abbreviations HU, SE, and CO should be included in the figure caption.

Line 588. Since this is an English text, the breed name should not be translated into Spanish. Instead, its original Galician name, Caldelá, should be used, as this is the region the breed originates from and the official name under which it is registered with the Ministry of Agriculture, Fisheries and Food.

https://www.mapa.gob.es/es/ganaderia/temas/zootecnia/razas-ganaderas/razas/catalogo-razas/bovino/caldela/default.aspx

https://razas.boaga.es/caldega/raza/historia/

Line 593. There is an extra space that should be removed.

The paragraphs between lines 578–583 and 612–615 contain repeated information. When reading the second paragraph, it feels like the same content has already been read. It would be advisable to revise the text to avoid such repetition.

Line 619. “th” should appear in superscript.

Lines 636–638. In this paragraph, "Sierra de Aracena" is mentioned both at the beginning and at the end, as if they were different places. This may be a mistake—perhaps another location was intended—or it may simply be a repetition. In any case, it should be corrected. If the repetition is intentional, it could be rephrased as: "…encompassing zones such as ... and the Sierra de Aracena itself."

Line 664. The figure number should be included.

Line 672. The figure number should be included.

Line 697. "Mendelian sampling variances" should be replaced with "MSVs".

Throughout the text, both "canonical discriminant analysis" and "discriminant canonical analysis" are used, which apparently refer to the same concept. If that is the case, I believe the same terminology should be used consistently to facilitate reader comprehension. Likewise, if they refer to the same concept, only one abbreviation should be used.

Line 838. The summary table of abbreviations should include those corresponding to Córdoba, Huelva, and Seville, which are currently missing.

Reference 23. The page numbers are missing; they should be 169–176.

Reference 38. The year should be in bold.

Table S1. For being consistent with all the ms, it should be replaced Sevilla by Seville.

Author Response

Comments and Suggestions for Authors

Line 76. The explanation of what F and f represent should be clearer, especially for readers without a technical background.

Response: Clarified in the body text.

Lines, 92, 95 and 128, and all other cases. It would be advisable to abbreviate 'nodal common ancestors' to maintain consistency with the rest of the text, which uses acronyms.

Response: We changed it across the document.

Lines 137–166. It is necessary to include the corresponding bibliographic references for these paragraphs. Although the content is widely known in the Spanish context, I believe that including the relevant citations could improve the text for international readers who wish to deepen their understanding of the topic.

Response: We added a reference to an article which makes a summary of all the data from the breed that is currently and officially available.

Line 232. This sentence must be corrected: When a NCA ...

Response: We corrected it.

Line 234. A space should be added in 'improvecomputation' to separate it into 'improve computation'.

Response: We added a space.

Line 246. The phrase "Mendelian sampling variances" should be removed, leaving only "MSVs," since the meaning of the abbreviation has already been explained earlier.

Reponse: We corrected and checked the rest of the text to maintain consitency.

Line 275. Final point should be added.

Response: Added.

Table 1. Since tables should be understandable on their own without requiring information from the main text, it would be better to write out CDA in its full form as Canonical Discriminant Analysis.

Response: Suggestion was followed.

Table 1. I suppose it should be 'Figure 1 and Supplementary Table S1'.

Response: Yes, we corrected it.

Figure 1 should be corrected in this sense: Figure 1. Map of Andalusia (Spain) ...

Response: Yes, we corrected it.

Line 341.  It would be advisable to abbreviate 'Mendelian Sampling Variances' to maintain consistency with the rest of the text, which uses acronyms. 

Response: We changed it across the document.

Line 346, 348 and 356. It would be advisable to abbreviate 'canonical discriminant analysis' to maintain consistency with the rest of the text, which uses acronyms. 

Response: We changed it across the document.

Line 427. The phrase "Mendelian sampling variances" should be removed, leaving only "MSVs," since the meaning of the abbreviation has already been explained earlier.

Response: We changed it.

Figure 3. The title for Figure 3 is missing and should be added below the figure.

Response. Added

Line 448. The phrase "Variance Inflation Factor" should be removed, leaving only "VIF," since the meaning of the abbreviation has already been explained earlier.

Response: Changed.

Line 526. The phrase "leave-one-out cross-validation" should be removed, leaving only "LOOCV," since the meaning of the abbreviation has already been explained earlier.

Response: We followed the reviewer’s suggestion.

Figure 4 and 5. In my opinion, it is not clear which part represents the prior classification and which represents the posterior classification.

Response: We clarified this as suggested.

Figure 5. The figure appears pixelated; I believe it would be advisable to modify it to improve the resolution and enlarge the text to enhance its readability and overall clarity.

Response: We improved figure quality.

Line 552. I believe this is the first time the abbreviation HU appears, so it should be written as Huelva (HU).

Response: We corrected it.

Line 552. I believe this is the first time the abbreviation SE appears, so it should be written as Seville (SE).

Response: We corrected it.

Line 559. I believe this is the first time the abbreviation CO appears, so it should be written as Córdoba (CO).

Response: We corrected it.

Figure 6. The meaning of the abbreviations HU, SE, and CO should be included in the figure caption.

Response: We corrected it.

Line 588. Since this is an English text, the breed name should not be translated into Spanish. Instead, its original Galician name, Caldelá, should be used, as this is the region the breed originates from and the official name under which it is registered with the Ministry of Agriculture, Fisheries and Food.

https://www.mapa.gob.es/es/ganaderia/temas/zootecnia/razas-ganaderas/razas/catalogo-razas/bovino/caldela/default.aspx

https://razas.boaga.es/caldega/raza/historia/

Response: We followed the reviewer suggestion.

Line 593. There is an extra space that should be removed.

Response: Removed.

The paragraphs between lines 578–583 and 612–615 contain repeated information. When reading the second paragraph, it feels like the same content has already been read. It would be advisable to revise the text to avoid such repetition.

Response: We apologize and revise the information to prevent repetition from occuring.

Line 619. “th” should appear in superscript.

Response: We followed teh reiewer’s suggestion.

Lines 636–638. In this paragraph, "Sierra de Aracena" is mentioned both at the beginning and at the end, as if they were different places. This may be a mistake—perhaps another location was intended—or it may simply be a repetition. In any case, it should be corrected. If the repetition is intentional, it could be rephrased as: "…encompassing zones such as ... and the Sierra de Aracena itself."

Response: We corrected it.

Line 664. The figure number should be included.

Response: We included it.

Line 672. The figure number should be included.

Response: We included it.

Line 697. "Mendelian sampling variances" should be replaced with "MSVs".

Response: We replaced it.

Throughout the text, both "canonical discriminant analysis" and "discriminant canonical analysis" are used, which apparently refer to the same concept. If that is the case, I believe the same terminology should be used consistently to facilitate reader comprehension. Likewise, if they refer to the same concept, only one abbreviation should be used.

Response: We clarified this and followe dreviewer suggestion.

Line 838. The summary table of abbreviations should include those corresponding to Córdoba, Huelva, and Seville, which are currently missing.

Response: We added them.

Reference 23. The page numbers are missing; they should be 169–176.

Response: Added.

Reference 38. The year should be in bold.

Response: Suggestion was followed.

Table S1. For being consistent with all the ms, it should be replaced Sevilla by Seville.

Response: We followed the reviewer’s suggestion.

Reviewer 2 Report

Comments and Suggestions for Authors

Review comments/Report

In the manuscript title Discriminant Canonical Mapping of the Origins in Andalusian 2 Black Cattle: Decomposition of Inbreeding and Coancestry through Mendelian Sampling Variances and Nodal Ancestors Contributions. The authors Macas et al. aim to identify the genetic origins and structure of the endangered Andalusian Black Cattle by decomposing inbreeding and ancestry into contributions from founders and nodal ancestors, using discriminant canonical analysis. This helps inform conservation strategies by highlighting key genetic contributors and regions of diversity. However, the presentation and expression of the content presented in their study is still lacking clarity in its expression. Therefore, some revision is still required before publication of this manuscript in Animals. The section wise comments are listed as;

Abstract and title

Line 1–5: The title is technically comprehensive but dense. It may be inaccessible to broader scientific readers. Consider simplifying the title to make it more approachable without losing specificity. Perhaps rephrase as "Mapping Genetic Origins and Diversity in Andalusian Black Cattle Through Canonical Analysis."

Line 31–44 (Abstract): The abstract effectively outlines the objectives and methods but suffers from wordiness and lack of focus on results. Streamline the abstract to more clearly present key findings and their implications, especially quantitative insights on genetic differentiation.

Simple Summary

Line 18–30: The summary is informative and well-written for a non-expert audience, though it repeats some concepts in the abstract.You might reduce redundancy by focusing more on the implications for conservation strategy in the summary.

Introduction

Line 76: some parameters explicitly use..." is grammatically awkward. Clarify by rephrasing to: "Some parameters explicitly incorporate the genetic contributions of individual ancestors, whereas others do not."

Line 88–93: The transition from theoretical metrics to nodal ancestor contributions is abrupt. Improve flow by briefly summarizing the need to move beyond traditional metrics like F and coancestry before introducing NCAs.

Materials and Methods

Line 186–192: The description of the dataset is clear but lacks rationale for the date ranges used. Briefly explain why January 1994 and May 2020 were chosen as bounds. Were they driven by data availability or biological events?

Line 204–209: Technical language like "ENDOG (v4.8)" is used without a short description. Provide a short clarification of what ENDOG and CFC software specifically calculate for clarity.

Line 259–273: Dense mathematical explanation without diagrams or simplification may hinder reader comprehension. Consider adding a figure or flowchart to visually demonstrate MSV decomposition steps.

Line 351–356: Method descriptions are thorough but overly reliant on software without elaborating on variable behavior. Include an explanation of why certain variables were expected to perform well in CDA based on theoretical expectations.

Results

Line 444–453 (3.2.1): VIF threshold values are applied correctly, but rationale for exclusion is not tied to variable interpretability. Mention how removing these variables impacts interpretability or reduces redundancy.

Line 529–533: Model underperformance in regions like Huelva and Jaén is noted but not deeply analyzed. Offer hypotheses on why the model fails in those regions—sampling bias, genetic homogeneity, etc.—and suggest further data collection.

Discussion

Line 569–577: The historical background is detailed, yet it repeats information from the introduction. Condense this to avoid redundancy and focus more on interpreting the implications of genetic patterns.

Line 722–727: The observation of retained diversity despite inbreeding is valuable but lacks support from concrete data. Reinforce this insight with a citation or quantified result from the results section.

Conclusion

Line 777–784: The conclusion is clear and aligns with study aims but could better highlight the novel methodological contribution. Emphasize how this approach can be generalized or applied to other breeds or conservation studies.

General comments

Many sections, especially methods, are heavy with technical jargon and equations. Incorporate more narrative explanation for complex formulas and analytical frameworks to support interdisciplinary accessibility.

Repetition between introduction, discussion, and background in results suggests that the manuscript would benefit from condensing. Aim for better cross-sectional integration, especially by merging overlapping content and emphasizing unique insights in each section.

Author Response

[Reviewer 2]

Comments and Suggestions for Authors

Review comments/Report

In the manuscript title Discriminant Canonical Mapping of the Origins in Andalusian 2 Black Cattle: Decomposition of Inbreeding and Coancestry through Mendelian Sampling Variances and Nodal Ancestors Contributions. The authors Macas et al. aim to identify the genetic origins and structure of the endangered Andalusian Black Cattle by decomposing inbreeding and ancestry into contributions from founders and nodal ancestors, using discriminant canonical analysis. This helps inform conservation strategies by highlighting key genetic contributors and regions of diversity. However, the presentation and expression of the content presented in their study is still lacking clarity in its expression. Therefore, some revision is still required before publication of this manuscript in Animals. The section wise comments are listed as;

Response. We thank the reviewer for his/her tiem and attention. We will describe and address how referee’ new recommendations were followed. A point-by-point response to comments is provided as well as a file where changes are highlighted.

Abstract and title

Line 1–5: The title is technically comprehensive but dense. It may be inaccessible to broader scientific readers. Consider simplifying the title to make it more approachable without losing specificity. Perhaps rephrase as "Mapping Genetic Origins and Diversity in Andalusian Black Cattle Through Canonical Analysis."

Response: We agree with the reviewer. However, we want the concepts of Mendelian Sampling Variances and Nodal Common Ancestors to appear on it. We shaped it to make it shorter.

Line 31–44 (Abstract): The abstract effectively outlines the objectives and methods but suffers from wordiness and lack of focus on results. Streamline the abstract to more clearly present key findings and their implications, especially quantitative insights on genetic differentiation.

Response: We followed the reviewer suggestion.

Simple Summary

Line 18–30: The summary is informative and well-written for a non-expert audience, though it repeats some concepts in the abstract.You might reduce redundancy by focusing more on the implications for conservation strategy in the summary.

 Response: We followed the reviewer suggestion.

Introduction

Line 76: some parameters explicitly use..." is grammatically awkward. Clarify by rephrasing to: "Some parameters explicitly incorporate the genetic contributions of individual ancestors, whereas others do not."

Response: We rewrote the paragraph according to the suggestion of both reviewers.

Line 88–93: The transition from theoretical metrics to nodal ancestor contributions is abrupt. Improve flow by briefly summarizing the need to move beyond traditional metrics like F and coancestry before introducing NCAs.

Response: We rewrote the Section according to the suggestion of both reviewers.

Materials and Methods

Line 186–192: The description of the dataset is clear but lacks rationale for the date ranges used. Briefly explain why January 1994 and May 2020 were chosen as bounds. Were they driven by data availability or biological events?

Response: We clarified it.

Line 204–209: Technical language like "ENDOG (v4.8)" is used without a short description. Provide a short clarification of what ENDOG and CFC software specifically calculate for clarity.

 Response: We followed the reviewer suggestion.

Line 259–273: Dense mathematical explanation without diagrams or simplification may hinder reader comprehension. Consider adding a figure or flowchart to visually demonstrate MSV decomposition steps.

Response: We followed the reviewer suggestion. Figure 1 was added and the rest renumbered.

Line 351–356: Method descriptions are thorough but overly reliant on software without elaborating on variable behavior. Include an explanation of why certain variables were expected to perform well in CDA based on theoretical expectations.

 Response: We followed the reviewer suggestion.

Results

Line 444–453 (3.2.1): VIF threshold values are applied correctly, but rationale for exclusion is not tied to variable interpretability. Mention how removing these variables impacts interpretability or reduces redundancy.

 Response: We followed the reviewer suggestion.

Line 529–533: Model underperformance in regions like Huelva and Jaén is noted but not deeply analyzed. Offer hypotheses on why the model fails in those regions—sampling bias, genetic homogeneity, etc.—and suggest further data collection.

Response: We followed the reviewer suggestion.

Discussion

Line 569–577: The historical background is detailed, yet it repeats information from the introduction. Condense this to avoid redundancy and focus more on interpreting the implications of genetic patterns.

 Response: We followed the reviewer suggestion.

Line 722–727: The observation of retained diversity despite inbreeding is valuable but lacks support from concrete data. Reinforce this insight with a citation or quantified result from the results section.

Response: We followed the reviewer suggestion.

Conclusion

Line 777–784: The conclusion is clear and aligns with study aims but could better highlight the novel methodological contribution. Emphasize how this approach can be generalized or applied to other breeds or conservation studies.

Response: We followed the reviewer suggestion.

General comments

Many sections, especially methods, are heavy with technical jargon and equations. Incorporate more narrative explanation for complex formulas and analytical frameworks to support interdisciplinary accessibility.

Repetition between introduction, discussion, and background in results suggests that the manuscript would benefit from condensing. Aim for better cross-sectional integration, especially by merging overlapping content and emphasizing unique insights in each section.

Response: We appreciate the reviewer’s insightful observation. In response, we have taken the following steps to improve clarity and reduce redundancy:
To make technical sections more accessible, we added a step-by-step narrative explanation of the Mendelian Sampling Variance (MSV) decomposition process. This walkthrough (now included as a new subsection) clearly describes each computational phase in non-technical language. Additionally, we included a visual flowchart (Figure 1) to assist readers from interdisciplinary backgrounds in understanding the analytical pipeline without relying solely on equations.
We reviewed the manuscript for overlap between the Introduction, Results, and Discussion. Repetitive historical and genetic context has been consolidated, with background now primarily housed in the Introduction. The Discussion has been revised to emphasize interpretation and synthesis of findings rather than reiteration. Where references to earlier material remain, we use cross-referencing to avoid duplication.

We believe these amendments significantly improve both the accessibility and cohesion of the manuscript and thank the reviewer again for the helpful suggestions.